# Consore: A Powerful Federated Data Mining Tool Driving a French Research Network to Accelerate Cancer Research

**DOI:** 10.3390/ijerph21020189

**Published:** 2024-02-07

**Authors:** Julien Guérin, Amine Nahid, Louis Tassy, Marc Deloger, François Bocquet, Simon Thézenas, Emmanuel Desandes, Marie-Cécile Le Deley, Xavier Durando, Anne Jaffré, Ikram Es-Saad, Hugo Crochet, Marie Le Morvan, François Lion, Judith Raimbourg, Oussama Khay, Franck Craynest, Alexia Giro, Yec’han Laizet, Aurélie Bertaut, Frederik Joly, Alain Livartowski, Pierre Heudel

**Affiliations:** 1Institut Curie, 75005 Paris, France; a-livartowski@unicancer.fr; 2Coexya, 69370 Saint-Didier-au-Mont-d’Or, France; nahidm@tcd.ie (A.N.); frederik.joly@coexya.eu (F.J.); 3Institut Paoli-Calmettes, 13009 Marseille, France; tassyl@ipc.unicancer.fr (L.T.); lemorvanm@ipc.unicancer.fr (M.L.M.); 4Gustave Roussy, 94805 Villejuif, France; marc.deloger@gustaveroussy.fr (M.D.); francois.lion@gustaveroussy.fr (F.L.); 5Data Factory & Analytics Department, Institut de Cancérologie de l’Ouest, 44805 Nantes-Angers, Francejudith.raimbourg@ico.unicancer.fr (J.R.); 6Institut Régional du Cancer de Montpellier, 34090 Montpellier, France; simon.thezenas@icm.unicancer.fr; 7Institut de Cancérologie de Lorraine, 54519 Nancy, France; e.desandes@nancy.unicancer.fr (E.D.); o.khay@nancy.unicancer.fr (O.K.); 8Centre Oscar Lambret, 59000 Lille, France; m-ledeley@o-lambret.fr (M.-C.L.D.); f-craynest@o-lambret.fr (F.C.); 9Centre Jean Perrin, 63011 Clermont Ferrand, France; xavier.durando@clermont.unicancer.fr (X.D.); alexia.giro@clermont.unicancer.fr (A.G.); 10Institut Bergonié, 33076 Bordeaux, France; a.jaffre@bordeaux.unicancer.fr (A.J.); y.laizet@bordeaux.unicancer.fr (Y.L.); 11Centre Georges Francois Leclerc, 21000 Dijon, France; icharifi@cgfl.fr (I.E.-S.); abertaut@cgfl.fr (A.B.); 12Centre Léon Bérard, 69008 Lyon, Francepierre-etienne.heudel@lyon.unicancer.fr (P.H.)

**Keywords:** cancer research, cancer, natural language processing, data mining, data warehouse, big data

## Abstract

Background: Real-world data (RWD) related to the health status and care of cancer patients reflect the ongoing medical practice, and their analysis yields essential real-world evidence. Advanced information technologies are vital for their collection, qualification, and reuse in research projects. Methods: UNICANCER, the French federation of comprehensive cancer centres, has innovated a unique research network: Consore. This potent federated tool enables the analysis of data from millions of cancer patients across eleven French hospitals. Results: Currently operational within eleven French cancer centres, Consore employs natural language processing to structure the therapeutic management data of approximately 1.3 million cancer patients. These data originate from their electronic medical records, encompassing about 65 million medical records. Thanks to the structured data, which are harmonized within a common data model, and its federated search tool, Consore can create patient cohorts based on patient or tumor characteristics, and treatment modalities. This ability to derive larger cohorts is particularly attractive when studying rare cancers. Conclusions: Consore serves as a tremendous data mining instrument that propels French cancer centres into the big data era. With its federated technical architecture and unique shared data model, Consore facilitates compliance with regulations and acceleration of cancer research projects.

## 1. Introduction

Cancer is a primary cause of mortality globally. It was responsible for close to 10 million deaths in 2020 [1]. This disease presents a myriad of unique pathologies [2], adding a layer of complexity to its diagnosis, research, and the improvement of care. The exploration of a specific pathology often demands the identification of a significant number of patients that typically exceeds the count available at a single healthcare facility [3]. Therefore, data integration and interoperability across institutions become essential to enhance cancer research and care.

Patient identification is a critical yet labor-intensive process [4]. The first step often requires manual review and interpretation of electronic health record (EHR) data, a process that is both slow and financially taxing. Despite this, it remains a necessary step, considering that approximately 80% of the pertinent clinical information is encapsulated within the text of health records [5]. To tackle these challenges, the contribution of natural language processing (NLP) techniques is a key asset to help physicians and data experts identify potential candidates for research projects.

Beyond being able to search available medical information, data must be structured and standardised to ensure their secondary reuse in research in accordance with the FAIR principles [6]. An initiative launched by eleven comprehensive cancer centres from the Unicancer network sought to enhance and expedite data sharing in oncology and has been involved in the creation of the OSIRIS model [7]. This common data model comprises a minimal set of clinical and genomic data, specific to oncology research, serving as the cornerstone of a larger initiative aimed at accelerating cancer research by simplifying the creation of cohorts of cancer patients with similar characteristics. Real-world data (RWD) in oncology is of paramount importance as it provides a comprehensive view of patient outcomes, treatment effectiveness, and cancer progression in diverse, real-world settings, thereby bridging the gap between clinical trials and everyday clinical practice and informing personalised treatment strategies and policy decisions.

To navigate regulatory hurdles and reduce data flow, the initiative implemented a federated technical architecture, thus avoiding the need for a single centralised data warehouse. However, this federated network requires a high level of harmonisation among centres to supply the common model. To this end, the French federation of comprehensive cancer centres (UNICANCER) developed Consore, a unique network equipped with a powerful federated search engine to revolutionise the utilisation of RWD. This tool enables the efficient and reliable identification of patient cohorts by digging into the electronic health records of millions of patients across eleven French comprehensive cancer centres.

Further to an exhaustive review of the clinical data warehousing landscape of the past years, we decided not to retain solutions requiring heavy infrastructures and prohibitive costs such as the caBIG initiative example [8] in the U.S in order to secure the participation of a large number of French comprehensive centres. Other solutions have been assessed such as i2b2 [9], transMART, and cBioPortal [10], but they revealed some shortcomings: poor data exploration and visualisation, not adapted to clinical data, and no NLP capabilities. This motivated the development of Consore, which stands out by offering an advanced NLP pipeline and a federated approach to seamlessly integrate and analyse diverse data sources across multiple centres.

In parallel, other French initiatives of clinical data warehouse implementation emerged like eHOP [11] or Dr. Warehouse [12], the latter more oriented toward narrative reports. Both tools focus on general medicine or rare diseases and offer advanced features for health data exploration but have not been designed specifically for cancer research. Both of these data warehouses are relying partially on a commercial database management system (Oracle™, Austin, TX, USA) that induces additional deployment and maintenance costs (Appendix A).

The project addresses four major challenges: (i) the aggregation of a tremendous amount of heterogeneous data; (ii) the semantic analysis of electronic health records, data standardisation, and modelling of the cancer disease; (iii) the technical implementation of a solution facilitating fast data querying at a national level; (iv) the development of ready-to-use services for clinicians and researchers. This paper presents an in-depth examination of each aspect of these challenges and their respective evaluations.

## 2. Materials and Methods

Overseen by UNICANCER, the federation of 18 French comprehensive cancer centres, the Consore project incorporates a multidisciplinary team of oncologists, bioinformaticians, project managers, data engineers, and Information Technology (IT) engineers. The implementation of Consore was authorised in November 2016 (N°2016-331) by the French regulatory authorities (Commission Nationale de l’Informatique et des Libertés, CNIL). Despite this approval, it remains essential to inform patients individually and collectively about the potential reuse of their health data for cancer research in compliance with European and French regulations.

### 2.1. Data Aggregation

Consore addresses various types of data coming from numerous disparate sources. This includes both structured and unstructured information in electronic medical records, demographic and administrative data, medical activity data derived from France’s nationwide diagnostic-related group (DRG)-based information system, hospital discharge data from the programme for medicalising information systems (PMSI), biobanking data, tumor characteristics, lab results, pharmaceutical data, and molecular alteration information.

### 2.2. Medical Concept Inference

Medical reports provide a treasure trove of data for Consore. However, harnessing these data presents a significant challenge due to the voluminous mass of raw unstructured data they contain. Given the content is natural language text, it poses a formidable task for AI to process. To surmount this challenge, Consore is equipped with a specialised web service for natural language processing (NLP) tasks. It receives medical reports, processes them via a spaCy NLP pipeline, and subsequently produces structured documents. This NLP pipeline carries out a series of operations on each report to extract as many medical concepts as possible. These operations include anonymizing personal information for regulatory reasons and text cleaning tasks such as stopword removal, tokenization, and lemmatization.

The web service extracts several concepts from the reports using NER models and structures them through entity-linking tasks. The latter are operated according to the same standards and classifications used for structured data sources in Consore, such as the following:− The “Classification Commune des Actes Médicaux” (CCAM) is the French classification for medical procedures [13];− The 10th revision of the International Classification of Diseases (ICD-10) [14] and the 3rd edition of the ICD for Oncology (ICD-O-3) [15]

Another issue we address using Consore’s NLP web service is redundancy and lack of precision. Indeed, for each patient, hundreds of reports are processed; most contain a redundant history section that is rarely updated. Moreover, in a single report, we tend to detect various occurrences of the same tumor, yet with different details: sometimes it is accompanied by the tumor location, its morphology, the associated biomarkers, the administered treatment, etc. These remain unitary concepts of different categories. Detecting them and representing them in a structured form does not solve all the challenges.

In fact, not only do we detect single concepts, but we also need to infer the links between them. For instance, there is a high interest in data structuring to find the dates linked to a concept in order to locate it in time. In addition to this example, we also tackle linking a treatment response to the involved treatment or recognising whether a tumor remains in its first stages or if a metastasis is diagnosed.

### 2.3. Information Consolidation Tool Using PMSI Data

The programme for medicalising information systems (PMSI) is a French mechanism of the national health system, aiming to reduce inequalities in resources between health institutions (Ordonnance of 24 April 1996). It stores quantified and standardised hospital discharge data to measure the activity and resources of healthcare institutions. The PMSI is thus a data source, which contains the patients’ stays in the healthcare facility with the reasons for hospitalisation, presented in a main diagnosis and associated diagnoses, both coded according to the ICD-10, and most treatments are coded according to the CCAM.

This data source plays a crucial role in Consore as it serves to validate identified diseases and treatments. Being of significant financial relevance to healthcare institutions, the PMSI provides a comprehensive dataset. However, Consore uses the PMSI to corroborate inferred medical information because the hospitalisation discharge data from the PMSI is suboptimal from a clinical perspective. While the data source is comprehensive, it lacks exhaustive documentation. For instance, for a given patient, we have the clinical diagnoses, but no information on the specific location or affected side, as well as the diagnosis date. Similarly, we can retrieve information on the occurrences of chemotherapy sessions, without any details on the administered medications.

### 2.4. Pivot Model for Diverse Data Sources

Consore handles a variety of data from several different sources that do not have a shared structure. Thus, a pivot model was designed in compliance with the OSIRIS model, capable of storing information from these diverse sources. This model addresses several hurdles such as partial data, conceptual links, volumetry and redundancy, and data sourcing.

− Partial data: we often deal with information about the morphology of a tumor, but we have no information about the initial diagnosis, its location, or treatment responses without the involved treatment.− Concept links: the pivot model must preserve, when applicable, the links between the detected concepts (e.g., the link between a metastasis and the primary tumor).− Volumetry and redundancy: Consore might detect or receive thousands of concepts, often redundant, for a single patient.− Data sourcing: for each data item, we need to identify its source and the date it was recorded in the information system.− In order to organise and classify all the identified concepts, we developed a common model defining the cancer disease based on several main hierarchical classes (or layers): cancers (all cancer recurrences for a given patient), tumor events (primary tumor, local, or metastatic relapse), acts (treatments and/or analysis), and documents (all the documents of a patient or available biological samples) (Figure 1).

Upon the completion of document processing through various pipelines, with each data source possessing its own, Consore carries out its primary function: data inference and structuring. The upcoming sections delve into metastasis structuring as an example of data structuring within Consore.

### 2.5. Data Inference and Structuring: An Illustration through Metastasis Structuring

The main asset of Consore lies in its ability to infer and structure data. For each patient, after processing and storing all their data in the pivot model, Consore uses the detected concepts to create a structured patient profile. This structuring process, involving the cleansing, merging, correcting, and selection of the most accurate data, results in a comprehensive document summarising the patient’s history. It enables the generation of a timeline detailing their condition, complete with events and observations since their initial examination. This structuring is complex, involving a series of rule executions in a specific order, with certain rules activated only under specific conditions.

Figure 2 shows the sequence of structuring rules leading to a patient profile within Consore’s inferred model. This model, similar to the elementary model, is solely used to store inferred data where concepts are connected to the patient at the core of the model. In contrast, in the elementary model, the core is a single document.

Here, we focus on the structuring rules for metastasis, a pivotal phase in cancer progression that is of great concern to oncology researchers. Identifying metastasis is technically challenging as it is often difficult to infer the primary tumor from which the metastasis arises.

The development of metastasis means in many cases the terminal stage of cancer. The primary task, therefore, is to accurately infer the starting date of the metastasis phase. However, as we deal with medical reports processed with NLP algorithms, a significant proportion of noise is introduced. These algorithms might pick up concepts of metastasis that are actually embedded in incorrect contexts such as a hypothesis or negation.

To address this challenge, we implemented a heuristic-based algorithm, summarised as follows:Retrieve all detected occurrences of the concepts “metastasis” and “relapses” that are located further from the primary tumor within the patient’s dataset. To maintain clarity, both types of occurrences will be referred to as “metastasis”.Sort these concepts based on their dates, either the date provided by the algorithm (e.g., “metastasis diagnosed on 24 May 2022”) or, in cases where no relevant date is found in the report, the date of the document itself.Determine the relevant date within the corpus of metastasis concepts. This involves defining a heuristic time interval (potentially 3/6 months or a year), starting from the first occurrence of a metastasis.Assign a weighting factor to each occurrence, considering the data source or the relevance of its associated date.Calculate the cumulative weight of the concepts falling within the defined interval, obtaining the interval’s total weight.If the interval’s weight exceeds a predefined empirical threshold, the start date within that interval is considered the commencement of the metastasis. If not, the process moves to the next interval, checks the same conditions, and repeats until a date is determined.

### 2.6. Data Processing and Structuring Pipeline

Consore’s design and implementation are tailored to meet the unique challenges of oncology data analysis. It is underpinned by a robust architecture that integrates advanced NLP techniques with probabilistic models and machine learning algorithms (Appendix A). This combination allows for effective processing, interpretation, and extraction of meaningful patterns from vast and diverse oncology datasets. The core of Consore is its advanced NLP pipeline, developed using spaCy, which processes EHRs to extract critical medical concepts. This pipeline employs convolutional neural networks (CNNs) and utilises FastText word vectors to accurately interpret the nuances of medical text. The pipeline stages include data anonymization, text cleaning (such as stopword removal, tokenization, and lemmatization), and entity recognition and linking, ensuring the transformation of raw text into structured, analysable data (Figure 3).

This pipeline begins with data ingestion, where raw data from various sources is collected. Following this, the data undergoes cleaning and standardisation to ensure consistency and quality. Next, our NLP algorithms annotate and extract relevant medical concepts from unstructured text. The structured data are then harmonised according to oncology-specific ontologies and stored in a searchable format, ready for analysis. Each of these steps is designed to ensure the integrity and utility of the data for research purposes (Figure 4).

### 2.7. The Role of Data Mining in Consore

Data mining plays a pivotal role in Consore’s ability to transform raw healthcare data into actionable insights. Through advanced data mining techniques, Consore can identify patterns, trends, and correlations within large datasets that would be impossible to discern manually.

### 2.8. Performance of the Consore Tool

We employed sophisticated data mining and NLP techniques to process and analyse the vast and complex datasets. Specifically, the results presented in this study are calculated using a combination of probabilistic models, semantic analysis, and machine learning algorithms tailored to understand and extract meaningful patterns from oncological data. The choice of these methods was driven by their proven effectiveness in similar contexts and their suitability for the specific challenges presented by cancer data. Recall, precision, and F1 scores were used to determine the Consore effectiveness in patients with unknown primary sites of cancer [16]. Recall was defined as the number of true positives (TP) over the number of true positives and false negatives (TP + FN), and precision as the number of true positives (TP) over the number of true positives and false positives (TP + FP). F1-score combined the two competing metrics as the harmonic mean of precision and recall (2 × Recall × Precision/(recall + precision)). An F1 score between 0.6 and 0.8 was considered as acceptable, a score between 0.8 and 0.9 as excellent, and more than 0.9 as outstanding [17].

## 3. Results

Consore is a software deployed within eleven major cancer centres in France (Figure 5).

The following table (Table 1) summarises the number of patients taken care of and whose data are included in Consore. This is to help represent the volume of data our software deals with.

The difference between the enumeration of patients and those with at least a cancer disease is due to the inclusion into the Consore of data of patients who have benign tumors or those who came into a centre for clinical tests that concluded an absence of cancerous disease.

### 3.1. Cancer of Unknown Primary

Cancers of unknown primary (CUP) are metastatic cancers for which a diagnostic work-up fails to identify the site of origin at the time of diagnosis and account for <5% of all cancers. It is a rare disease and often poorly documented in the EHR and for which the diagnosis is made by default, that is to say, when all the primary cancer sites have been eliminated, which can sometimes take several months.

### 3.2. Results at the “Centre Léon Bérard” (CLB)

After analysis of an “institutional” database and the Consore query, we have a total of 145 CUP at the “Centre Leon Berard”. The selection criteria used in Consore were as follows: patients initially diagnosed with de novo metastatic cancer and for whom the mention “unknown primary” was present in the electronic health record since 1 January 2010. The difficulty of this query lies in the fact that it is based both on a textual search and also on data structured by Consore (the dates of diagnosis and the metastatic stage). Consore offers data for 2577 patients, with 1 patient from the institutional database not found by Consore. Focusing only on cancer diagnoses between 1 January 2019 and 30 June 2021, we manually reviewed 121 electronic medical records. Concerning these 121 patients, 69 had a CUP while 52 patients had cancer whose primary was identified. The large number of false positives is linked to the period of diagnostic uncertainty which can last several months with cancers of unknown primary (the diagnosis of cancer of unknown primary being ultimately excluded when a primary tumor is identified). So, over this period, Consore found 69 additional diagnoses but detected at the same time 52 false positives.

To assess the performance, recall, precision, and F1 scores were calculated as follows:o Recall = 99%;o Precision = 57%;o F1-score = 0.66.o Calculation of inverse recall is not possible because manual control of all EMRs to identify true negatives is not possible.

### 3.3. Results at the “Institut Curie” (IC)

Based on the Consore query first designed at the “Centre Léon Bérard”, we identified a selection of 133 CUP at Institut Curie. The primary comparison with the institutional TransCUPtomics [18] study (48 CUP patients) showed the identification of only eight patients (17%). This poor result is mainly due to the initial criteria: in the TransCUPtomics study, 15 patients were diagnosed before 2010 and 33 patients had a metastatic relapse after the initial diagnosis. However, there was also a lack of keywords to properly identify CUP patients in Consore. In the second step, we optimised the Consore query by adding the following list of French keywords: “ACUP”, “primitif inconnu”, “de primitif inconnu”, “sans primitif retrouve”, “sans primitif connu”, “d’origine indéterminée”, “pas de primitif retrouve”, “d’origine inconnue”, “recherche de la tumeur primitive”, and “autre primitif” (Figure 6).

With this new query, we identified 2871 CUP patients on the overall database where we retrieved 45 of the 48 TransCUPtomics patients (94%): 2 patients remained not found because of a lack of EHRs (less than five documents available), 1 patient had a CUP status unclear with minimal description in the EHR. In order to assess the positive predictive value (PPV), we reviewed a sample of 119 patients diagnosed between 1 January 2019 and 30 June 2021. Concerning these 119 patients, 58 had a CUP while 26 patients had cancer whose primary was identified (false positives linked to the period of diagnostic uncertainty). So, over this period, Consore found 58 additional diagnoses but detected at the same time 61 false positives.

To assess the performance, recall, precision, and F1 scores were calculated as follows:o Recall = 94%;o Precision = 56%;o F1-score = 0.7.o Calculation of inverse recall is not possible because manual control of all EMRs to identify true negatives is not possible.

## 4. Discussion

Considering the growing impact of RWD on clinical practice and research, the development of digital solutions to gather and use such data becomes essential. Beyond the description of this innovative tool, our work underlines the major role of collaborative efforts in constructing a federated technical architecture and agreeing on a unified data model.

With the creation of the OSIRIS model, French cancer centres have laid down a foundation of a minimal data set, which serves as an essential starting point for any collaborative cancer research. Consore, to date, has facilitated numerous research projects on large cohorts of cancer patients, including timely studies on topics such as immunotherapy or COVID-19 [19]. The tool’s role in RWD research has been bolstered through collaboration with the Health Data Hub [20] and through a project initiative aimed at cross-referencing Consore-provided data with data from the national health data system [21]. A dialogue has also been initiated to foster communication between this data model and international initiatives, such as OMOP [22].

Numerous French and international consortiums are aiming to structure information within medical records to facilitate their reuse for research purposes. Examples include Dr Warehouse and the EHOP warehouse in France, and CancerLinq [23] and FLATIRON [24] in the United States. In this ecosystem, Consore stands out as one of the few solutions dedicated to oncology with a federated architecture, employing natural language processing.

However, this project is not without limitations. While the federated architecture is a strength, it also brings some issues. Each cancer centre maintains its own EHR, which contains heterogeneous data sources in different formats and varying data quality levels. Although the OSIRIS model allows data format standardisation, quality variability remains a challenge that must be acknowledged, quantified, and considered in the analyses and outcomes of multicentre projects. The question of completeness and quality of the data source as well as the results persist for each research work.

Another identified limitation deals with the common data model, which relies on primary tumor identification. While suitable for most solid tumors, such as breast or lung cancer, it presents more challenges and therefore errors for haematological cancers, sarcomas, and skin cancers. For these types of neoplastic diseases, the histological type is indeed more critical than the primary tumor location. Efforts are underway to improve these results.

As an expert tool, Consore is dedicated to oncology and its NLP algorithms have been exclusively trained on French language-based medical reports. A wider dissemination of Consore at an international level or for other chronic diseases (such as chronic obstructive pulmonary disease or chronic renal failure) would require substantial modifications such as the development of an English version or significant data model transformations.

The evolution of Consore is an ongoing journey. We are exploring enhancements such as incorporating more advanced machine learning models (from CNN to LLM such as BERT and GPT), expanding our data sources to include genomic and radiotherapy dosimetry data, and improving our algorithms for even better performance and accuracy. We also plan to extend our collaborations internationally, aiming to make Consore a versatile tool for global cancer research communities.

Recent advancements in NLP have significantly enhanced the ability to extract and analyse complex information from EHRs, a critical component of healthcare research and delivery. As outlined in various studies, the application of advanced neural networks and deep learning approaches, such as those discussed by Assale et al., has considerably outperformed traditional methods, enabling more accurate interpretation of unstructured medical text [25]. The development of large language models like GatorTron demonstrates the potential of NLP to improve clinical tasks and data analysis significantly, as evidenced by improvements in various NLP tasks [26]. Moreover, the systematic review of clinical text data in machine learning highlights the capability of NLP to unlock valuable clinical narratives for extensive analysis [27], while other studies emphasise the suitability of various deep learning algorithms for EHR applications [28]. A recent study illustrated the potential of advanced NLP in clinical pharmacology. This highlights the technology’s capability to enhance data extraction and analysis, underscoring the transformative impact NLP could have on clinical trial methodologies and drug development efficiency [29]. Together, these advancements underscore the growing importance and effectiveness of NLP in enhancing the accessibility and utility of healthcare data, thereby supporting more informed and efficient research and clinical decision making.

The previously highlighted limitations suggest several potential areas for future enhancement. Special emphasis will be placed on improving the comprehensiveness and quality of the results, as well as bridging the data model into the OMOP-CDM format (Observational Medical Outcomes Partnership—Common Data Model) to facilitate international engagement. The pivotal aspect, thus far, revolves around the update on data quality. Despite our awareness of the inherent constraints of Real-World Data (RWD), we have established a dedicated working group to systematically assess this quality and enhance the overall robustness of the results.

## 5. Conclusions

Consore stands as a powerful tool dedicated to oncology and is capable of modelling patients’ neoplastic histories by structuring data from EHRs. Despite the discussed limitations, Consore has already demonstrated its potential in accelerating the identification of patient cohorts and the implementation of research projects. With its federated architecture and the use of a common data model, Consore is playing a key role in the development of multicentric projects in the field of oncology at a national level. The next steps will aim to enhance interoperability between Consore and other international networks as well as the development of next-generation NLP algorithms based on large language models (LLM) [30].

## Figures and Tables

**Figure 1 ijerph-21-00189-f001:**
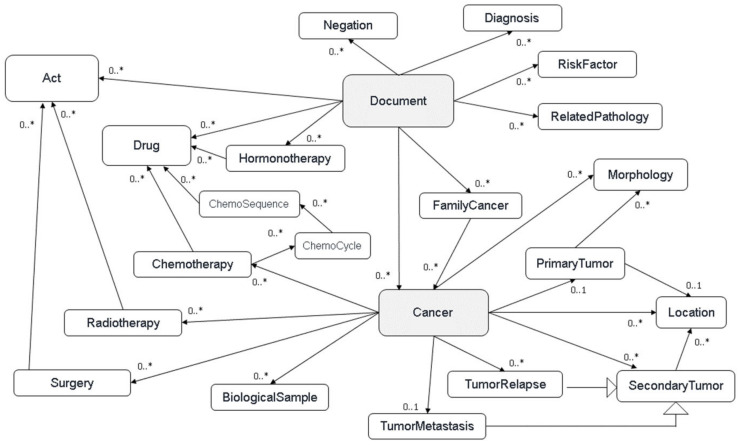
Consore’s elementary data model.

**Figure 2 ijerph-21-00189-f002:**
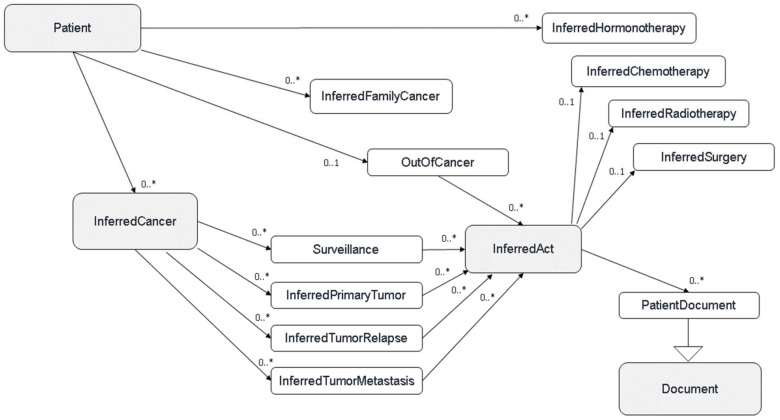
Consore’s inferred model.

**Figure 3 ijerph-21-00189-f003:**
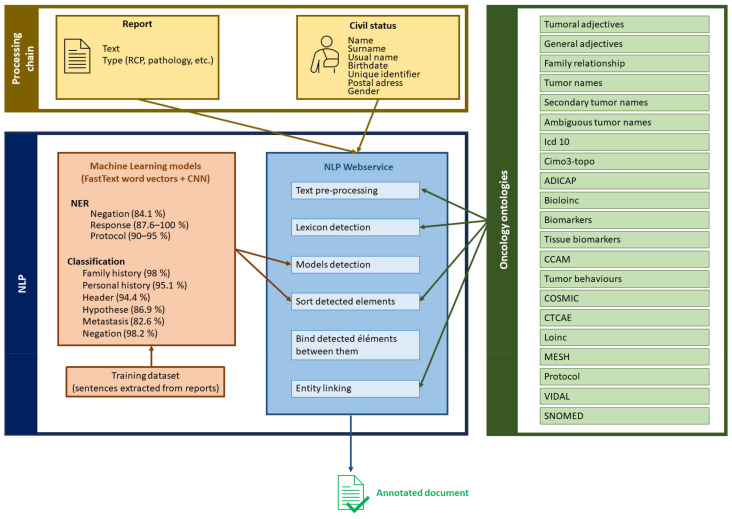
The Consore NLP architecture.

**Figure 4 ijerph-21-00189-f004:**
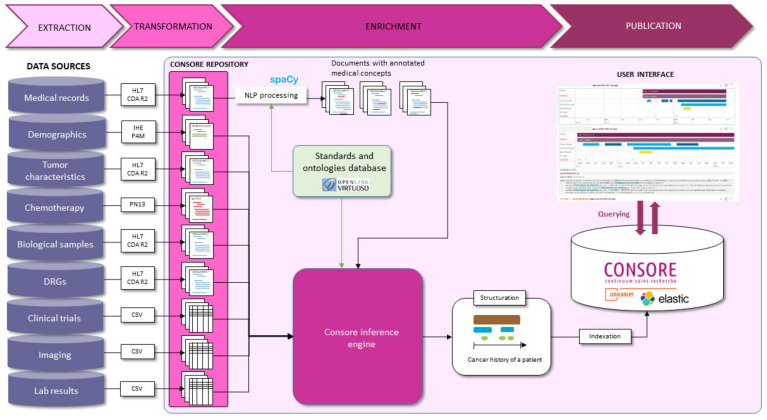
The Consore data processing pipeline (ETEP).

**Figure 5 ijerph-21-00189-f005:**
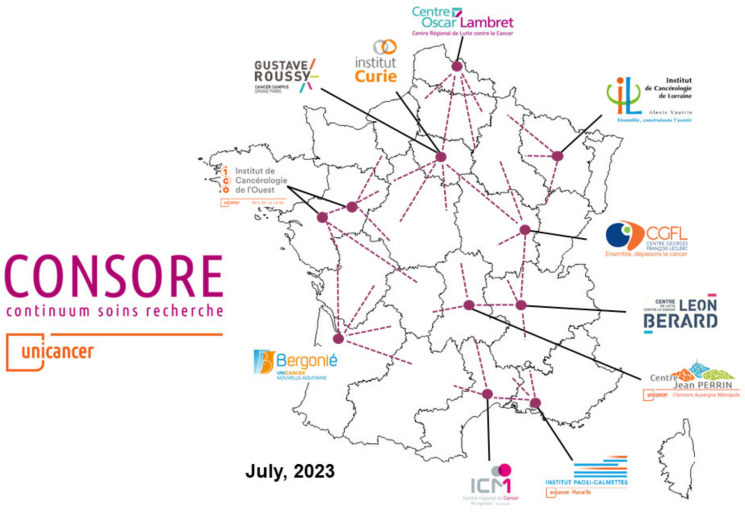
The Consore network in France (July 2023).

**Figure 6 ijerph-21-00189-f006:**
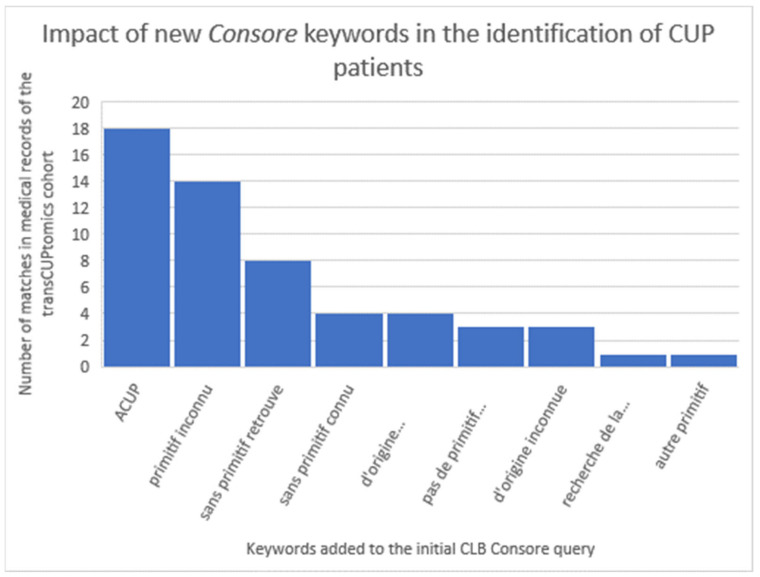
Impact of new Consore keywords in the identification of CUP patients at the “Institut Curie” (based on the initial query provided by the “Centre Léon Bérard”).

**Table 1 ijerph-21-00189-t001:** Overview of data volumes in the different Consore data warehouses (* February 2023; ** August 2023).

French Cancer Centres	Nb of Patients	Nb of Patients with at Least One Cancer	Nb of Patients with a Metastatic Relapse	Nb of Medical Records
Institut Curie *	572,421	280,924	95,025	13,431,874
Centre Léon Bérard *	359,634	207,657	85,210	18,711,561
Institut Paoli-Calmettes *	347,415	136,500	43,767	4,464,580
Gustave Roussy *	399,665	237,132	96,074	12,856,023
Institut de Cancérologie de l’Ouest	N/A (deployment in progress)	N/A	N/A	
Centre Oscar Lambret *	182,436	118,506	57,784	5,865,404
Institut du Cancer de Montpellier *	176,257	79,601	34,138	3,401,825
Centre Georges-François Leclerc *	282,948	79,592	36,635	3,207,721
Centre Jean Perrin *	397,179	124,080	44,548	2,776,005
Institut Bergonié *	285,129	153,589	52,290	3,806,476
Institut de Cancérologie de Lorraine **	247,869	63,350	19,105	1,096,485

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
