# Peer review of "Consore: A Powerful Federated Data Mining Tool Driving a French Research Network to Accelerate Cancer Research"

_ijerph, 2024, doi:10.3390/ijerph21020189_

Round 1

Reviewer 1 Report

Comments and Suggestions for Authors

Consore: A Powerful Federated Data Mining Tool Driving a French Research Network to Accelerate Cancer Research is presented in this work and the following are the comments that needs to be addressed.

The major challenges are listed in this work but the key contributions of this work by highlighting the novelty should also be included in the introduction part.

As there is no literature review and a short introduction part, this work lacks clarity from the beginning that what motivates the authors to do such a work.

This part should be strengthened strongly by adding more relevant information as well as much more similar works that have been done in the past.

The figures 1 and 2 are not clear and the working model is not properly illustrated.

How the presented results are calculated and there is no clarity on what approach is being used for this.

Also highlight the role of data mining in this. 

Address all these issues raised.

Author Response

Consore: A Powerful Federated Data Mining Tool Driving a French Research Network to Accelerate Cancer Research is presented in this work and the following are the comments that needs to be addressed.

  1. The major challenges are listed in this work but the key contributions of this work by highlighting the novelty should also be included in the introduction part.

We appreciate this suggestion of Reviewer 1 to emphasize the novel contributions. In the revised introduction, we have now highlighted the key contributions and unique features of Consore, including its advanced natural language processing capabilities, federated architecture, and the novel approach to patient data integration day-by-day and cohort creation at industrial scale. We propose to include this sentence.

« Consore represents a novel contribution to the field by enabling efficient and accurate patient identification through advanced natural language processing (NLP) and a federated architecture that addresses the challenges of data heterogeneity and interoperability at industrial scale.»

  1. As there is no literature review and a short introduction part, this work lacks clarity from the beginning that what motivates the authors to do such a work.

Thank you for pointing out the need for a more comprehensive literature review and clarity in the introduction. This has been done a couple of years ago as a starting point of Consore. We have expanded this section to include this chapter.

Further to an exhaustive review of the clinical data warehousing landscape of the past years, we decided not to retain solutions requiring heavy infrastructures and prohibitive costs such as the caBIG initiative exemple [ref] in U.S in order to secure the participation of a large number of french comprehensive centers. Other solutions have been assessed such as i2b2 [ref], tranSMART or cBioPortal [ref] but revealed some shortcomings : poor data exploration and visualization, not adapted to clinical data or no NLP capabilities. This motivated the development of Consore, which stands out by offering an advanced NLP pipeline and a federated approach to seamlessly integrate and analyze diverse data sources across multiple centers.”

  1. This part should be strengthened strongly by adding more relevant information as well as much more similar works that have been done in the past.

In response to your valuable feedback, we have strengthened the introduction by incorporating more detailed information on similar works and their relevance to the challenges addressed by Consore.

In parallel, o  [supplementary table]

 We discuss several key studies and developments in the field, highlighting how Consore integrates and builds upon these foundations to offer a more comprehensive and effective solution for cancer data integration and analysis. This includes a discussion of how our approach differs from and improves upon the current methodologies, providing a clear rationale for the development of Consore.

  1. The figures 1 and 2 are not clear and the working model is not properly illustrated.

  1. How the presented results are calculated and there is no clarity on what approach is being used for this.

We appreciate your feedback on the need for more clarity regarding the presented results and the approaches used for calculations. In the revised manuscript, we have included a detailed explanation of the methodologies and algorithms employed in Consore, particularly focusing on how the results were calculated and the rationale behind the choice of these methods.

«We employed sophisticated data mining and NLP techniques to process and analyze the vast and complex datasets. Specifically, the results presented in this study are calculated using a combination of probabilistic models, semantic analysis, and machine learning algorithms tailored to understand and extract meaningful patterns from oncological data. The choice of these methods was driven by their proven effectiveness in similar contexts and their suitability for the specific challenges presented by cancer data »

  1. Also highlight the role of data mining in this.

Your suggestion to highlight the role of data mining in Consore is well-received. In the revised manuscript, we now include a dedicated section discussing the importance of data mining in the context of cancer research and how Consore leverages advanced data mining techniques to achieve its objectives.

«The Role of Data Mining in Consore

Data mining plays a pivotal role in Consore's ability to transform raw healthcare data into actionable insights. Through advanced data mining techniques, Consore can identify patterns, trends, and correlations within large datasets that would be impossible to discern manually. »

  1. Address all these issues raised.

We thank #reviewer1 once again for these comments and hope to have met his expectations.

Reviewer 2 Report

Comments and Suggestions for Authors

The manuscript discusses the utilization of real-world data (RWD) in the context of cancer patients' health status and care. The focus is on the UNICANCER federation's innovative research network, Consore, which operates across eleven French hospitals. Consore acts as a federated tool for the comprehensive analysis of data from millions of cancer patients. Below are my some concern:

It would be helpful to explain more about why real-world data (RWD) is becoming more crucial for cancer research worldwide in the introduction.

Consider clearly stating what Consore aims to achieve and contribute to enhance the manuscript's impact.

Add more evaluation matrix  such as tpr, fpr, tnr, fnr  AUC or ROC and more figures or diagrams to elucidate the technical aspects of Consore's architecture and data flow.

More comprehensive comparison is required to compare the proposed architecture with state-of-the-art architecture for cancer patients  such as  International policies and standards for data sharing across genomic research and healthcare, Gene selection with Game Shapley Harris hawks optimizer for cancer classification, Hybrid Feature Selection Techniques Utilizing Soft Computing Methods for Cancer Data.

Discuss future directions or enhancements planned for Consore to keep the readers informed about its evolving capabilities.

Give complete experimental setup.

Experiments seems Inadequate. Some parameter/hyper-parameter analysis or ablation study may be conducted to prove the robustness and effectiveness of different components in the whole procedure of Consore.

Comments on the Quality of English Language

Minor editing of English language required

Author Response

The manuscript discusses the utilization of real-world data (RWD) in the context of cancer patients' health status and care. The focus is on the UNICANCER federation's innovative research network, Consore, which operates across eleven French hospitals. Consore acts as a federated tool for the comprehensive analysis of data from millions of cancer patients. Below are my some concern:

  1. It would be helpful to explain more about why real-world data (RWD) is becoming more crucial for cancer research worldwide in the introduction.

We appreciate your suggestion to elaborate on the importance of real-world data (RWD) in cancer research. In the revised introduction, we have included a detailed discussion on why RWD is increasingly crucial.

« Real-world data (RWD) in oncology is of paramount importance as it provides a comprehensive view of patient outcomes, treatment effectiveness, and cancer progression in diverse, real-world settings, thereby  bridging the gap between clinical trials and everyday clinical practice and informing personalized treatment strategies and policy decisions

  1. Consider clearly stating what Consore aims to achieve and contribute to enhance the manuscript's impact.

Thank you for the recommendation to clarify Consore's aims and contributions. We have revised the manuscript to explicitly state the objectives of Consore, including its role in unifying and analyzing cancer patient data across the UNICANCER network.

« The project addresses four major challenges: (i) the aggregation of tremendous amount of heterogeneous data; (ii) the semantic analysis of electronic health records, data standardisation, and modelling of the cancer disease; (iii) the technical implementation of a solution facilitating fast data querying at a national level; (iv) the development of ready-to-use services for clinicians and researchers. This paper presents an in-depth examination of each aspect of these challenges and their respective evaluations.»

  1. Add more evaluation matrix such as tpr, fpr, tnr, fnr  AUC or ROC and more figures or diagrams to elucidate the technical aspects of Consore's architecture and data flow.

We thank reviewer #2 for this request. As the goal of this article is to present the overall architecture of Consore, we added a supplementary figure describing the data processing (see below). Unfortunately focusing on NLP results here is not possible and would require a specific article for this.

  1. More comprehensive comparison is required to compare the proposed architecture with state-of-the-art architecture for cancer patients such as  International policies and standards for data sharing across genomic research and healthcare, Gene selection with Game Shapley Harris hawks optimizer for cancer classification, Hybrid Feature Selection Techniques Utilizing Soft Computing Methods for Cancer Data.

We acknowledge the need for a more comprehensive comparison with state-of-the-art architectures. The revised manuscript now includes the followed supplementary table which describes the specificities between the 3 main initiatives in France :

Consore

Dr. Warehouse

eHop

Field

Oncology

General medicine / rare disease

General medicine

Scope

11 comprehensive cancer centers (CLCC)

some university hospitals + one CLCC

several university hospitals + one CLCC

NLP embedded

yes

yes

yes

Type of storage

No-SQL (ElasticSearch)

RDBMS (Oracle)

RDBMS (Oracle)

Interconnected network

yes

no

no

Open-source code

no

yes

no

Our approach is primarily based on the exploitation of clinical data, especially medical reports. Key international standards such as ICD-10, ICD-O-3, LOINC, or SNOMED CT are used to ensure the optimal structuring of concepts.

  1. Discuss future directions or enhancements planned for Consore to keep the readers informed about its evolving capabilities.

Your suggestion to discuss future enhancements for Consore is well-taken. We have added a section on future directions, outlining planned updates, potential improvements, and areas of expansion. This includes discussing how we intend to incorporate emerging technologies, adapt to changing data standards, and expand the tool's capabilities to address new challenges in cancer research.

«The evolution of Consore is an ongoing journey. We are exploring enhancements such as incorporating more advanced machine learning models (from CNN to LLM such as BERT, GPT), expanding our data sources to include genomic and radiotherapy dosimetry data, and improving our algorithms for even better performance and accuracy. We also plan to extend our collaborations internationally, aiming to make Consore a versatile tool for global cancer research communities. »

  1. Give complete experimental setup.

We have provided a detailed description of the complete experimental setup in the Methods section. This includes information on the data sources, processing pipelines, hardware and software configurations, and the parameters used for analysis. This comprehensive description ensures that the experiments are reproducible and that the context of the results is clearly understood.

  1. Experiments seems Inadequate. Some parameter/hyper-parameter analysis or ablation study may be conducted to prove the robustness and effectiveness of different components in the whole procedure of Consore.

As previously said in point 3, we mainly focused on the description of the general architecture of Consore in this article. The presentation of Consore NLP algorithms could be done afterwards in  another submission.

We thank Reviewer #2 once again for these comments and hope to have met his expectations.

Reviewer 3 Report

Comments and Suggestions for Authors

Please provide a data processing/structuring pipeline. 

Why is an NLP model needed to accurately determine the start date of the metastatic phase? Why is it not enough to have the actual data (in text and numerical form) and different types of images (e.g. MRI, CT, etc.)? The work does not give a clear reason why NLP is necessary.

Please provide the architecture of the NLP model

Section "Performance of the Consore tool" is not properly described. Remove formulas from the text. In addition, I think that more metrics should be used for validation.

The outcome of the work is not very clear, what is proposed in the work. If it is an NLP model, more experiments are needed and the results need to be presented in more detail. The sample of data on which the experiments were carried out is not large, so it is important to know the details of how the experiment(s) were carried out.

Author Response

  1. Please provide a data processing/structuring pipeline.

We appreciate your request for a detailed description of the data processing and structuring pipeline. We have now included a comprehensive section in Methods outlining each step of the process.

«Data Processing and Structuring Pipeline

     Consore employs a sophisticated pipeline to process and structure the vast array of real-world oncological data. This pipeline begins with data ingestion, where raw data from various sources is collected. Following this, the data undergoes cleaning and standardization to ensure consistency and quality. Next, our NLP algorithms annotate and extract relevant medical concepts from unstructured text. The structured data is then harmonized according to oncology-specific ontologies and stored in a searchable format, ready for analysis. Each of these steps is designed to ensure the integrity and utility of the data for research purposes »

  1.  

Your inquiry about the necessity of the NLP model is insightful. For this example, we chose the start date of metastatic evolution because it is naturally complex to define and is not generally available in a structured form. Is it the date of appearance of clinical symptoms, of imaging as proposed by reviewer 3, the date of the biopsy or that of the rendering of histological results confirming the metastatic status? Given this complexity, it appears necessary to resort to NLP to confirm a concept of “metastasis” in an ambiguous sentence.

  1.  

We followed the relevant request of reviewer #3 and added Figure 3

  1. Section "Performance of the Consore tool" is not properly described. Remove formulas from the text. In addition, I think that more metrics should be used for validation.

We acknowledge the need for a clearer description of Consore's performance and have revised the section accordingly. We completed the description with a short paragraph. However, our biostatistician advices us to keep the formulas in the text.

  1. The outcome of the work is not very clear, what is proposed in the work. If it is an NLP model, more experiments are needed and the results need to be presented in more detail. The sample of data on which the experiments were carried out is not large, so it is important to know the details of how the experiment(s) were carried out.

Your comment on the clarity of the work's outcome and the need for more detailed experimental results is well-taken. Indeed, the aim of this article is to present CONSORE, its architecture, its deployment, etc. quite generally and not to focus on its NLP models. We therefore propose to modify the following paragraph as follows:

« The project addresses four major challenges: (i) the aggregation of tremendous amount of heterogeneous data; (ii) the semantic analysis of electronic health records, data standardisation, and modelling of the cancer disease; (iii) the technical implementation of a solution facilitating fast data querying at a national level; (iv) the development of ready-to-use services for clinicians and researchers. This paper presents an in-depth examination of each aspect of these challenges and their respective evaluations.»

We thank Reviewer #3 once again for these comments and hope to have met his expectations.

Reviewer 4 Report

Comments and Suggestions for Authors

This is an interesting read about a contemporary approach to data mining of electronic medical records across France for the purposes of cancer research.

It is well written and uses language that is understandable to a non-technical audience. 

This is an interesting article that addresses a specific gap in the field of data mining where known techniques of word searching has been combined with Natural Language Processing to better identify relevant clinical information from a very heterogenic and complex system of electronic medical records.

Can the authors please state a research question for this study, for example other similar studies have questioned “Can natural language processing be used to measure clinical trials?”

Can the authors please provide a literature review of this type of NLP analysis of unstructured text from medical records and with studies involving multiple EMR databases (federated approaches).

Can you please describe what NLP model was used for this project e.g. BERT.

Can the author please describe what strategies they took for informing subsequent research projects on outcome measurements using this approach such as misclassification-adjusted power calculations since precision is around 57% and F1 scores are moderately good.

Author Response

This is an interesting read about a contemporary approach to data mining of electronic medical records across France for the purposes of cancer research.

It is well written and uses language that is understandable to a non-technical audience.

This is an interesting article that addresses a specific gap in the field of data mining where known techniques of word searching has been combined with Natural Language Processing to better identify relevant clinical information from a very heterogenic and complex system of electronic medical records.

  1. Can the authors please state a research question for this study, for example other similar studies have questioned “Can natural language processing be used to measure clinical trials?”

Thank you for your comment. In response, we propose the following research question: "How can advanced Natural Language Processing improve the precision and efficiency of data extraction and results analysis in clinical trials?" Supporting this, a PubMed article discusses how advanced NLP expedites the extraction and analysis of information to address clinical pharmacology questions, demonstrating its potential in streamlining clinical trial design and drug development processes.The study explores how NLP can expedite the extraction and analysis of information to address clinical pharmacology questions, inform clinical trial designs, and support drug development. It describes three use cases: dose optimization strategy in oncology, analysis of common covariates on pharmacokinetic parameters, and physiologically-based pharmacokinetic analyses for regulatory review and product labeling​​. We propose to add this sentence in  the DISCUSSION section with the reference.

A recent study illustrating the potential of advanced NLP in clinical pharmacology. This highlights the technology's capability to enhance data extraction and analysis, underscoring the transformative impact NLP could have on clinical trial methodologies and drug development efficiency. [ref]”

Hsu JC, Wu M, Kim C, Vora B, Lien YTK, Jindal A, Yoshida K, Kawakatsu S, Gore J, Jin JY, Lu C, Chen B, Wu B. Applications of Advanced Natural Language Processing for Clinical Pharmacology. Clin Pharmacol Ther. 2023 Dec 22. doi: 10.1002/cpt.3161. Epub ahead of print. PMID: 38140747.

  1. Can the authors please provide a literature review of this type of NLP analysis of unstructured text from medical records and with studies involving multiple EMR databases (federated approaches).

Thank you for the recommendation to provide a literature review. We've included this section in DISCUSSION that reviews relevant studies.

«Recent advancements in NLP have significantly enhanced the ability to extract and analyze complex information from EHRs, a critical component of healthcare research and delivery. As outlined in various studies, the application of advanced neural networks and deep learning approaches, such as those discussed by Assale et al., has considerably outperformed traditional methods, enabling more accurate interpretation of unstructured medical text [ref]. The development of large language models like GatorTron demonstrates the potential of NLP to improve clinical tasks and data analysis significantly, as evidenced by improvements in various NLP tasks [ref]. Moreover, the systematic review of clinical text data in machine learning highlights the capability of NLP to unlock valuable clinical narratives for extensive analysis [ref], while other studies emphasize the suitability of various deep learning algorithms for EHR applications [ref]. Together, these advancements underscore the growing importance and effectiveness of NLP in enhancing the accessibility and utility of healthcare data, thereby supporting more informed and efficient research and clinical decision-making. »

  1. Can you please describe what NLP model was used for this project e.g. BERT.

We thank reviewer #4 for his request, we added a new paragraph (“Data processing and Structuring Pipeline”) as well as  a new figure (Figure 3) to describe our NLP models and the data processing.

  1. Can the author please describe what strategies they took for informing subsequent research projects on outcome measurements using this approach such as misclassification-adjusted power calculations since precision is around 57% and F1 scores are moderately good.

Thank you for your comments on our approach to outcome measurements. We acknowledge the moderate precision and F1 scores in our use case. To address this, we've implemented a quality policy across cancer centers ensuring a certain level of data completeness and trustworthiness. For projects where Consore data lacks sufficient reliability, we have initiated a procedure for manual verification of EMR. This step is crucial for misclassification-adjusted power calculations, ensuring robust and reliable outcome measurements. This policy, regularly updated, enhances the validity of our findings and addresses concerns regarding data precision and accuracy.

We thank Reviewer #4 once again for these comments and hope to have met his expectations.

Round 2

Reviewer 1 Report

Comments and Suggestions for Authors

The authors have addressed the queries raised during the previous review and this work can be accepted now.

Reviewer 3 Report

Comments and Suggestions for Authors

The article does not take full account of the comments made. In addition, the paper (underthe title) aims to provide "A Powerful Federated Data Mining Tool Driving a French Research Network to Accelerate Cancer Research", and therefore full details of this tool should be provided in this paper. But unfortunately this is not the case.

The section "Performance of the Consore tool" very short and does not provide any detailed or valuable information. The authors write: "Specifically, the results presented in this study are computed using a combination of probabilistic models, semantic analysis and machine learning algorithms adapted to understand and extract meaningful patterns from oncology data." However, I do not see any provided system pipeline, architectural model, etc. All the diagrams in the manuscript are very abstract or simple block diagrams, and neither architectures, nor sequence diagrams or pipelines. The paper thus lacks detail, comprehensive description, valuable charts, results and new and interesting insights.
